# Nuclear Factor-κB Signaling Regulates the Nociceptin Receptor but Not Nociceptin Itself

**DOI:** 10.3390/cells13242111

**Published:** 2024-12-20

**Authors:** Lan Zhang, Ulrike M. Stamer, Robin Moolan-Vadackumchery, Frank Stüber

**Affiliations:** 1Department of Anaesthesiology and Pain Medicine, Inselspital, Bern University Hospital, University of Bern, 3010 Bern, Switzerland; ulrike.stamer@unibe.ch (U.M.S.); frank.stueber@insel.ch (F.S.); 2Department for BioMedical Research, University of Bern, 3008 Bern, Switzerland; 3Graduate School for Cellular and Biomedical Sciences, University of Bern, 3012 Bern, Switzerland

**Keywords:** cell cultures, cytokines, NFκB, nociceptin, nociceptin receptor, signal transduction

## Abstract

The nociceptin receptor (NOP) and nociceptin are involved in the pathways of pain and inflammation. The potent role of nuclear factor-κB (NFκB) in the modulation of tumor necrosis factor-α (TNF-α) and interleukin (IL)-1β on the nociceptin system in human THP-1 cells under inflammatory conditions were investigated. Cells were stimulated without/with phorbol-myristate-acetate (PMA), TNF-α, IL-1β, or PMA combined with individual cytokines. To examine NFκB’s contribution to the regulation of the nociceptin system, PMA-stimulated cells were treated with NFκB inhibitor BAY 11-7082, JSH-23, or anacardic acid before culturing with TNF-α or IL-1β. *NOP* and prepronociceptin (*ppNOC*) mRNA were quantified by RT-qPCR; cell membrane NOP and intracellular nociceptin protein levels were measured by flow cytometry. Phosphorylation and localization of NFκB/p65 were determined using ImageStream. PMA + TNF-α decreased *NOP* mRNA compared to stimulation with PMA alone, while PMA + IL-1β did not. BAY 11-7082 and JSH-23 reversed the repression of *NOP* by PMA + TNF-α. TNF-α and IL-1β attenuated PMA’s upregulating effects on *ppNOC*. None of the inhibitors preserved the upregulation of *ppNOC* in PMA + TNF-α and PMA + IL-1β cultures. TNF-α strongly mediated the nuclear translocation of NFκB/p65 in PMA-treated cells, while IL-1β did not. Proinflammatory cytokines suppressed *NOP* and *ppNOC* mRNA in PMA-induced human THP-1 cells. NFκB signaling seems to be an important regulator controlling the transcription of NOP. These findings suggest that the nociceptin system may play an anti-inflammatory role during immune responses.

## 1. Introduction

Nociceptin and the nociceptin receptor (NOP) are expressed in human peripheral blood leukocytes and regulated in various diseases [1,2]. Emerging preclinical and clinical studies point to the nociceptin system as a therapeutic target in the treatment of pain and inflammatory diseases [3,4,5,6,7].

In previous studies, phorbol-myristate-acetate (PMA) induced a strong upregulation of nociceptin in human monocytic cells, as well as in peripheral blood leukocytes [2,8,9]. Furthermore, lipopolysaccharide (LPS) abolished the upregulation of nociceptin induced by PMA in human monocytic THP-1 cells [8], a cell line derived from the peripheral blood of an acute monocytic leukemia patient. However, the mechanisms involved in the antagonistic effect of LPS on PMA-induced nociceptin expression are still not clear.

LPS activates Toll-like receptor 4 (TLR4) signaling and subsequently increases nuclear factor-κB (NFκB) phosphorylation, a major transcription factor complex which is related to the transcriptional regulation of proinflammatory cytokines such as tumor necrosis factor-α (TNF-α) and interleukin (IL)-1β [10,11,12]. TNF-α and IL-1β are key cytokines produced predominantly by monocytes and macrophages during the early phase of LPS stimulation [13]. In addition, previous studies have reported that the accumulation of activated NFκB in PMA-differentiated THP-1 cells primes the cells for markedly increased sensitivity to LPS and led to a rapid secretion of TNF-α and IL-1β following LPS challenge [14,15]. Moreover, it has been revealed that TNF-α triggers the nuclear translocation of the NFκB/RelA(p65) subunit promoting the transcription of responsive genes, which is similar to that of LPS [16,17]. In a previous study, TNF-α and IL-1β exerted a suppressive effect on *NOP,* as well as on *ppNOC* mRNA, in human peripheral blood cells [18].

We therefore hypothesized that inflammatory cytokines play a crucial role in the modulation of nociceptin and its receptor under inflammatory conditions and that NFκB signaling participates in the cytokine-mediated regulation of the nociceptin system.

## 2. Materials and Methods

### 2.1. Cell Cultures

THP-1 cells (CLS, Eppelheim, Germany) were maintained in RPMI-1640 medium supplemented with 2 mM L-glutamine, 100 U/mL penicillin, 100 µg/mL streptomycin, and 10% fetal calf serum (FCS) at 37 °C and 5% CO_2_ atmosphere (all from Sigma-Aldrich, Buchs, Switzerland). Cells were seeded at a density of 1.5 × 10^5^ cells per well in 0.5 mL culture medium for 48-well culture plates and 1.2 × 10^6^ cells per well in 4 mL culture medium for 6-well culture plates (TPP, Trasadingen, Switzerland).

#### 2.1.1. Dose–Response Experiments

Dose-dependent effects of TNF-α, IL-1β, IL-10 on *NOP* and *ppNOC* mRNA expression, as well as the inhibitory effects of NFκB inhibitors (BAY 11-7082 10 nM–1 µM, JSH-23 100 nM–10 µM, and anacardic acid 100 nM–10 µM), were evaluated. Based on these pilot experiments, cytokines at a concentration of 10 ng/mL and NFκB inhibitors at a concentration of 100 nM were used in the subsequent experiments (Sigma-Aldrich, Buchs, Switzerland).

#### 2.1.2. Co-Stimulation of THP-1 with PMA and Cytokines

To examine the effects of cytokines on *NOP* and *ppNOC* mRNA, cells were cultured without/with PMA 5 ng/mL and without/with TNF-α 10 ng/mL or IL-1β 10 ng/mL for 24 h. PMA at a concentration of 5 ng/mL was chosen based on our previous studies [2,9].

#### 2.1.3. Inhibition Experiments

To further investigate the contribution of NFκB signaling to the regulation of *NOP* and *ppNOC* mRNA, cells were stimulated without/with PMA 5 ng/mL for 24 h and then treated without/with one of the NFκB inhibitors (BAY 11-7082 100 nM, JSH-23 100 nM, and anacardic acid 100 nM) for 1 h prior to culturing without/with TNF-α 10 ng/mL or IL-1β 10 ng/mL for 6 and for 12 h.

### 2.2. RNA Isolation, cDNA Synthesis, and Relative Quantification

Total RNA was extracted from cells using a high pure RNA isolation kit (Roche, Rotkreuz, Switzerland), and the quality and concentration of RNA were detected using a NanoDrop 2000 (Thermo Scientific, Reinach, Switzerland). Subsequently, cDNA was synthesized (Transcriptor High Fidelity cDNA Synthesis Kit, Roche, Rotkreuz, Switzerland).

The mRNA expression of *NOP* (TaqMan probes ID Hs00173471_m1), *ppNOC* (TaqMan probes ID Hs00918595_m1), *IL1B* (TaqMan probes ID Hs01555410_m1), and two reference genes, hypoxanthine phosphoribosyl-transferase 1 (*HPRT1*) (TaqMan probes ID Hs02800695_m1) and glyceraldehyde-3-phosphate dehydrogenase (*GAPDH*) (TaqMan probes ID Hs02758991_g1), were determined using TaqMan gene expression assays (Thermo Scientific, Reinach, Switzerland). Real-time quantitative PCR (qPCR) was conducted in 384-well plates by a LightCycler^®^ 480 (Roche, Rotkreuz, Switzerland) using 5 µL of TaqMan Fast Advanced Master Mix (2×), and 0.5 µL TaqMan Assay in a final volume of 10 µL. All samples were measured in duplicate. cDNA from SK-N-DZ cells, a human neuroblastoma cell line, was used as a calibrator.

Standard curves were generated separately for each gene of interest, as well as for each reference gene using serial dilutions of cDNA templates. mRNA expression was analyzed using the advanced relative quantification analysis of the LightCycler^®^ 480 software 1.5.0. *NOP* and *ppNOC* mRNA levels were computed based on qPCR amplification efficiency and the threshold cycle values and calculated as target/reference ratio of each sample normalized to the calibrator used in the PCR reactions.

### 2.3. Flow Cytometry

#### 2.3.1. Measurement of Cell Membrane NOP

Cell surface staining of NOP protein was performed as previously described [8]. Briefly, cells were collected, washed with ice-cold PBS, and suspended in 10% human serum AB type (Sigma-Aldrich, Buchs, Switzerland) for 20 min on ice to block unwanted FC receptor-involved staining. Cells were then incubated with 5 µg/mL mouse anti-human NOP mAb (Sigma-Aldrich, Buchs, Switzerland) or an isotype-control antibody (BD Biosciences, Allschwil, Switzerland) in 50 µL hypotonic saponin solution for 5 min on ice [19]. After washing with perm buffer, cells were incubated with anti-human NOP mAb 5 µg/mL or isotype-control antibody 5 µg/mL (BD Biosciences, Allschwil, Switzerland) for one hour on ice and washed. Subsequently, cells were stained with 1 μg/mL PE-conjugated goat anti-mouse secondary antibody (Thermo Scientific, Reinach, Switzerland) for one hour on ice in the dark, washed, and fixed with 1% paraformaldehyde (Sigma-Aldrich, Buchs, Switzerland).

#### 2.3.2. Measurement of Intracellular Nociceptin

To stain intracellular nociceptin, cells were fixed and permeabilized (BD Cytofix/CytopermTM Kit, BD Biosciences, Allschwil, Switzerland) and incubated with rabbit anti-human nociceptin antibody (Phoenix Pharmaceuticals, Karlsruhe, Germany) or isotype-control antibody (Abcam, Cambridge, UK) at a final concentration of 5 μg/mL for one hour at room temperature (RT). After washing three times, cells were stained with 1 μg/mL PE-conjugated mouse anti-rabbit secondary antibody (BD Biosciences, Allschwil, Switzerland) for one hour at RT in the dark, washed, and suspended in staining buffer.

Samples were measured on a CytoFLEX S Flow Cytometer within 4 h. At least 10,000 events in gated single cell population were recorded for each sample. Median fluorescence intensity of NOP and nociceptin was calculated using the FlowJo V10.8.1 software (TreeStar Inc., Ashland, OR, USA).

#### 2.3.3. Measurement of Phosphorylated NFκB/p65

Nuclear translocation of phosphorylated NFκB/p65 was assessed using ImageStream technology. Briefly, 1.2 × 10^6^ cells were seeded per well in 4 mL culture medium in a 6-well plate. Cells were stimulated without/with PMA 5 ng/mL for 24 h, followed by treatment without/with TNF-α 10 ng/mL or IL-1β 10 ng/mL for 1 h. Cells were then directly fixed with 4% formaldehyde for 15 min at 37 °C, collected and washed with 1× tris-buffered saline (TBS), permeabilized with 0.5% Triton X-100 for 20 min at RT, and blocked unwanted background with 3% goat serum (all from Sigma-Aldrich, Buchs, Switzerland). Subsequently, cells were incubated overnight with anti-NFκB/p65 antibody or isotype control antibody at 5 µg/mL at 4 °C. After incubation, samples were washed three times and incubated with Alexa Flour 594-tagged secondary antibody at 2.5 µg/mL for one hour at RT in the dark. Afterwards, samples were washed twice and nuclei were counterstained with DAPI (Sigma-Aldrich, Buchs, Switzerland). NFκB phosphorylation and nuclear translocation were measured on a Cytek^®^ Amnis^®^ ImageStream^®X^ Mk II imaging flow cytometer. A total of 5000 events in single-cell gating were recorded for each sample. Image analysis was performed and similarity scores were calculated using IDEAS^®^ software 6.3 to quantify the translocation of NFκB/p65 (Cytek^®^ Biosciences, Amsterdam, The Netherland).

### 2.4. Statistical Analysis

Statistical analysis was performed using STATISTICA 13.0 (StatSoft, Inc., Tulsa, OK, USA). Data are presented as medians, interquartile ranges, 10–90 percentiles, and means. Normality was determined using the Shapiro–Wilk test. Kruskal–Wallis, Mann–Whitney U, and Wilcoxon tests were applied for data with non-normal distribution. Results were corrected for multiple testing. *p* < 0.05 was considered statistically significant.

## 3. Results

### 3.1. Modulation of NOP and Nociceptin by PMA

NOP was constitutively expressed in THP-1 cells at both the mRNA and protein levels. Although *NOP* mRNA was not affected by PMA after 24 h, cell surface NOP protein levels were increased in PMA-treated cells (Figure 1A). *ppNOC* mRNA was below the detection limit in untreated cells, whereas intracellular nociceptin protein could be detected. In THP-1 cells stimulated with PMA 5 ng/mL for 24 h, *ppNOC* mRNA and intracellular nociceptin protein levels were increased (Figure 1B).

### 3.2. Dose-Dependent Effects of Cytokines and NFκB Inhibitors

Concentration ranges of 0.1 to 10 ng/mL for TNF-α and IL-1β and 1 to 100 ng/mL for IL-10 were used. Results showed that TNF-α combined with PMA suppressed *NOP* and *ppNOC* mRNA expression in a dose-dependent manner. IL-1β dose-dependently prevented the upregulating effect of PMA on *ppNOC* mRNA after 24 h. No obvious dose–response effects of IL-10 on *ppNOC* and *NOP* mRNA levels were observed (Figure 2A,B). Based on these results, TNF-α 10 ng/mL and IL-1β 10 ng/mL were used in further experiments. In addition, cells were pretreated without/with various concentrations of NFκB inhibitors BAY 11-7082 (BAY), JSH-23 (JSH), or anacardic acid (AA) for 1 h, followed by exposure to LPS 100 ng/mL for 6 h. qPCR analysis indicated BAY, JSH, and AA affected the LPS-induced *IL1B* mRNA expression in a dose-dependent manner (Figure 2C). Based on this pilot testing, NFκB inhibitors were used at 100 nM in the subsequent experiments.

### 3.3. Effects of TNF-α and IL-1β on NOP and ppNOC mRNA

To investigate the effects of cytokines on *NOP* and *ppNOC* mRNA expression, cells were cultured without/with PMA 5 ng/mL in the presence or absence of TNF-α or IL-1β for 24 h. The results showed that PMA alone had no impact on *NOP* mRNA, whereas PMA + TNF-α exerted a suppressive effect on *NOP* mRNA expression compared to PMA-treated cultures, while IL-1β combined with PMA did not affect *NOP* mRNA levels (Figure 3A). As for nociceptin, PMA significantly increased *ppNOC* mRNA expression after 24 h, compared to an untreated control. TNF-α and IL-1β attenuated the upregulating effect of PMA on *ppNOC* mRNA levels compared to a PMA-treated group. No changes in *ppNOC* mRNA were observed in cells treated with TNF-α or IL-1β alone (Figure 3B).

### 3.4. Effects of NFκB Inhibitors on the Regulation of NOP and ppNOC mRNA

To further examine the potential role of NFκB signaling in regulating the nociceptin system, three specific NFκB inhibitors were employed (BAY 11-7082, a κB kinase inhibitor; JSH-23, a NFκB/p65 nuclear translocation inhibitor; and anacardic acid, a histone acetyltransferase inhibitor). qPCR analysis demonstrated that co-stimulation with PMA and TNF-α for 6 h resulted in a decrease in *NOP* mRNA levels compared to stimulation with PMA alone. This downregulation was abolished by BAY and partially reversed by JSH. *NOP* mRNA levels in the PMA + BAY + TNF-α and in the PMA + JSH + TNF-α were increased 66.3 (33.0–166.2)% and 32.8 (8.7–82.1)%, respectively, compared to PMA + TNF-α samples. A trend of antagonistic effects was also observed in the PMA + AA + TNF-α samples after 12 h; however, the differences did not reach statistical significance. In addition, these NFκB inhibitors had no influence on the expression of *NOP* mRNA in PMA + IL-1β samples (Figure 4A–C).

With regard to nociceptin, it is noteworthy that none of the NFκB inhibitors had any effect on the expression of *ppNOC* mRNA in both PMA + TNF-α and PMA + IL-1β cultures (Figure 4D–F).

### 3.5. Nuclear Translocation of NFκB/p65

NFκB/p65 and nuclear signals were acquired using ImageStream, and the similarity score was calculated for NFκB/p65 nuclear localization (Figure 5A–C). Image analysis revealed that TNF-α 10 ng/mL increased the intensity of Alexa 594-labeled anti-NFκB/p65 (green) and strongly induced NFκB/p65 nuclear translocation in THP-1 cells after one hour compared to untreated controls. Weak activation of NFκB/p65 was observed in cells treated with IL-1β 10 ng/mL; no NFκB/p65 nuclear translocation was detected. THP-1 cells stimulated with PMA 5 ng/mL for 24 h, NFκB/p65 signal was increased compared to the untreated control. However, phosphorylated NFκB was cytosolic and accumulated around the nucleus in PMA-treated cells. Strong translocation of cytoplasmic NFκB/p65 to the nuclei was visible through imaging as green spots inside the nuclei after PMA-induced THP-1 cell exposure to TNF-α for one hour, whereas only a weak NFκB/p65 nuclear localization was observed in the PMA + IL-1β samples.

## 4. Discussion

The present study reveals that TNF-α combined with PMA suppressed *NOP* mRNA levels in THP-1 cells. TNF-α and IL-1β attenuated the upregulating effect of PMA on *ppNOC* mRNA expression. NFκB signaling contributed to the transcriptional control of the nociception receptor NOP but not nociceptin itself.

The constitutive expression of NOP in human peripheral blood leukocytes and the release of nociceptin by these cells under inflammatory conditions indicate that the nociceptin system is involved in immune responses [1,2,20]. In addition, elevated plasma nociceptin levels and aberrant NOP expression in blood cells of patients suffering from pain or inflammation provide further support for the nociceptin system as a potential target for the treatment of inflammatory-related diseases [4,5,6,7,21,22]. However, mechanisms contributing to the regulation of nociceptin and its receptor are not fully revealed.

In previous studies, PMA was the only substance with a strong upregulating effect on nociceptin in human monocytic cells (MM6 and THP-1) and blood leukocytes [2,8,9]. Moreover, Toll-like receptor (TLR) agonists repressed nociceptin mRNA, as well as protein levels induced by PMA in THP-1 cells [8]. Mechanisms underlying these antagonizing effects exerted by TLR agonists on nociceptin, however, are still not clear.

LPS and PMA have different impacts on cytokine profiles in THP-1 cells. TLR4 signaling pathways culminate in the activation of NFκB, which controls a large array of inflammatory cytokine gene expression [23,24]. PMA-induced THP-1 cells are more sensitive to LPS, resulting in higher release of inflammatory cytokines upon LPS stimulation. TNF-α and IL-1β are the most prominent proinflammatory cytokines released by macrophages following LPS stimulation [15,25]. In addition, in blood cells stimulated with LPS, TNF-α and IL-1β contributed to the reduction in *NOP* mRNA levels [18]. Accordingly, it can be speculated that the abundant secretion of TNF-α and IL-1β triggered by LPS from the PMA-induced THP-1 cells may suppress the mRNA expression of nociceptin and NOP.

### 4.1. ppNOC, NOP mRNA, and Cytokines

The present qPCR analysis indicates that while the proinflammatory cytokines TNF-α and IL-1β repressed the upregulating effect of PMA on *ppNOC* mRNA expression, the anti-inflammatory cytokine IL-10 did not. As for NOP, PMA alone had no effect on its mRNA level, while the combination of PMA + TNF-α suppressed *NOP* mRNA expression. This is in line with our previous findings that TNF-α decreased *NOP* mRNA level in human blood leukocytes. While IL-1β attenuated *NOP* mRNA expression in blood cells, it did not impact the level of *NOP* mRNA in THP-1 cells [18]. A reason for this difference might be that primary cells are more sensitive to the stimulus than cell lines. Of note, in whole-blood cultures, crosstalk between leukocyte subsets may influence cellular response to external stimuli, which may consequently comprehensively impact *NOP* mRNA levels. It has been well documented that IL-1β and TNF-α are two principal cytokines that promote NFκB activation [26]. In addition, it was previously observed that mitogen-activated protein kinases, but not NFκB, contributed to the induction of *ppNOC* mRNA in MM6 cells and in blood leukocytes [9]. Therefore, the potential role of NFκB signaling in the regulation of TNF-α and IL-1β on *NOP* and *ppNOC* mRNA expression in PMA-induced THP-1 cells was investigated in this study.

### 4.2. NFκB Regulates NOP mRNA

NFκB is one of the main transcription factors whose modulation triggers a cascade of signaling events and plays critical roles in multiple physiological and pathological processes, including pain and inflammation. The activation kinetics of NFκB phosphorylation is highly dynamic and the translocation of NFκB/p65 from cytoplasm to nucleus is critical for NFκB-dependent gene expression. LPS, cytokines, and PMA are known activators of NFκB signaling. Compared to LPS and cytokines, PMA weakly induces NFκB DNA binding [27]. In order to determine the contribution of NFκB signaling in the regulation of the nociceptin system, different specific NFκB inhibitors BAY 11-7082, JSH-23, and anacardic acid—which, respectively, block κB kinase, nuclear translocation, and histone acetyltransferase activity—were used in the present investigation. qPCR results indicated that BAY and JSH reversed the repression of *NOP* mRNA expression induced by PMA + TNF-α in THP-1 cells after 6 h. A trend of an antagonistic effect by AA was observed after 12 h, but it did not reach statistical significance. As for nociceptin, none of these NFκB inhibitors affected *ppNOC* mRNA levels in PMA + TNF-α and in PMA + IL-1β samples. Nevertheless, this confirms previous findings that NFκB signaling did not contribute to the upregulating effects of PMA on nociceptin in human MM6 cells, as well as in blood leukocytes.

In addition, the present ImageStream results demonstrate that NFκB accumulated in the cytoplasm of THP-1 cells in the presence of PMA 5 ng/mL for 24 h. TNF-α promoted the nuclear translocation of NFκB in these cells. This is consistent with results from previous reports indicating that phosphorylated NFκB induced by PMA in THP-1 cells accumulated in the cytoplasm during the differentiation process [14]. Interestingly, the nuclear translocation of NFκB did not occur in PMA + IL-1β samples. One possible explanation might be that TNF-α and IL-1β differentially influence NFκB activity, which may lead to distinct effects on cell responses. Although TNF-α and IL-1β activate NFκB, differences exist, i.e., TNF-α uniquely activates caspase-8 pathway [28], IL-1β activates hypoxia-inducible factor-1 [29]. A complex signaling network is formed when the cells response to a particular stimulus and can lead to numerous cellular responses. On the other hand, the current ImageStream data could provide a reason for the cause of differential effects of IL-1β and TNF-α on *NOP* mRNA expression in PMA-induced THP-1 cells.

In clinical settings, increased plasma nociceptin and reduced *NOP* mRNA have been detected in patients with sepsis. In addition, previous studies revealed that nociceptin activated NFκB [30,31]. In this respect, a high extracellular nociceptin level may act as a negative regulator of the nociceptin receptor, preventing an overflow of NOP signaling. Mechanisms underlying the suppressive effects of TNF-α and IL-1β on nociceptin under inflammatory conditions remain unclear at this point. Yet, the inhibitory effects of proinflammatory cytokines on *NOP* and *ppNOC* mRNA levels together with previous findings that nociceptin suppressed mRNA expression of *TLR*s and counteracted proliferation, migration and the increasing IL-1β mRNA levels by LPS via TLR4 in human U87 cells [8,32], suggest anti-inflammatory activity of the nociceptin system during immune responses.

Some limitations of the present study need to be considered. First, although THP-1 is a widely used model for investigating monocyte–macrophage biology, the cultures may not accurately reflect the regulation of the nociceptin system in blood cells under pathophysiological conditions and in vivo. Second, this study mainly assessed the regulatory effects of TNF-α and IL-1β on *NOP* and *ppNOC* mRNA expression and focused on the possible participation of the NFκB signal transduction pathway in the nociceptin system. However, the inflammatory factors involved in the pathological process of pain and inflammation are diverse and complex and affect multiple signaling pathways and their downstream targets. Effects of TNF-α and IL-1β on the phosphorylation and degradation of the inhibitor of κB protein, a key regulator of the NFκB pathway, in PMA-treated THP-1 cells need to be investigated in a future study.

## 5. Conclusions

Proinflammatory cytokines repressed *NOP* and *ppNOC* mRNA in PMA-induced human monocytic THP-1 cells. The NFκB signaling pathway seems to play a pivotal role in controlling the transcription of the nociceptin receptor.

## Figures and Tables

**Figure 1 cells-13-02111-f001:**
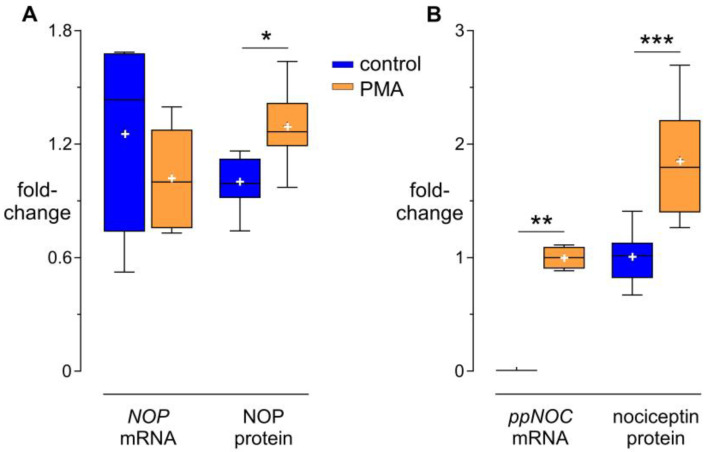
Effects of PMA on NOP (**A**) and nociceptin (**B**) mRNA and protein levels in THP-1 cells. Cells were treated with PMA 5 ng/mL or without PMA (controls) for 24 h. Median with interquartile range and 10–90 percentiles; mean “+”; mRNA and protein data are representative of six and twelve independent experiments, respectively. Wilcoxon tests * *p* < 0.05; ** *p* < 0.005; *** *p* < 0.001, compared to the respective controls.

**Figure 2 cells-13-02111-f002:**
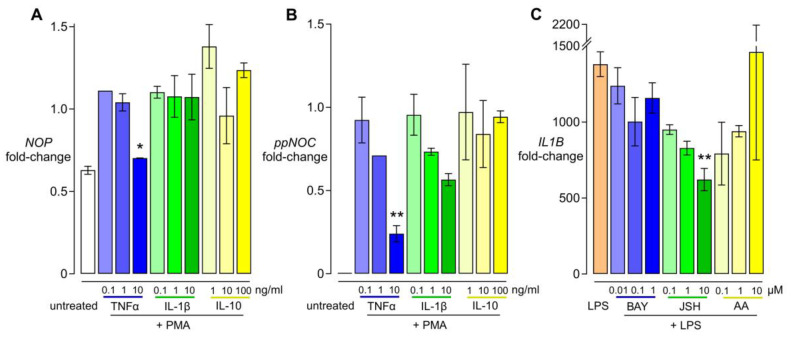
Dose-dependent effects of cytokines and NFκB inhibitors. THP-1 cells were cultured without/with PMA 5 ng/mL and without/with various concentrations of individual cytokines for 24 h. mRNA expression of *NOP* (**A**) and *ppNOC* (**B**) is presented as mRNA ratio related to the respective PMA samples. (**C**) Dose-dependent inhibitory effects of different NFκB inhibitors on *IL1B* mRNA levels induced by LPS. Cells were pre-treated without/with various concentrations of BAY 11-7082 (BAY), JSH-23 (JSH), or anarcadic acid (AA) for 1 h prior to culturing without/with LPS 100 ng/mL for 6 h. *IL1B* mRNA levels are presented as mRNA ratio related to an untreated group. Data are from two independent experiments and measures are expressed in mean with range. One-sample t test, * *p* < 0.05; ** *p* < 0.005, compared to the PMA group (*NOP* and *ppNOC*) and compared to the LPS group (*IL1B*).

**Figure 3 cells-13-02111-f003:**
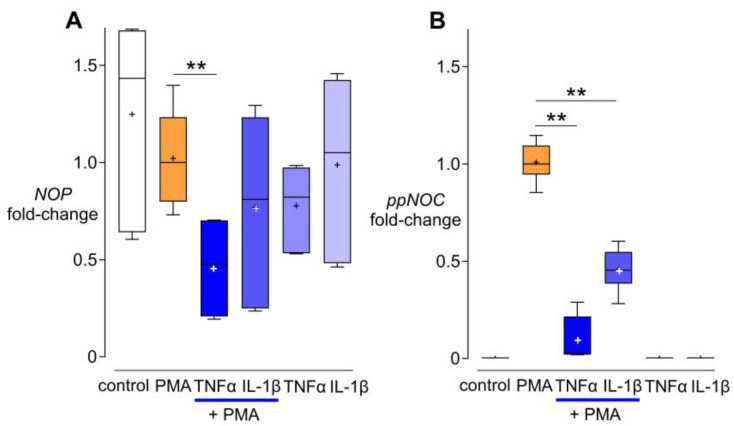
Effects of TNF-α and IL-1β on *NOP* and *ppNOC* mRNA expression. THP-1 cells were cultured without/with PMA 5 ng/mL in the presence or absence of TNF-α 10 ng/mL or IL-1β 10 ng/mL for 24 h. Quantitative PCR analysis of *NOP* (**A**) and *ppNOC* mRNA expression (**B**). Fold change values in mRNA levels are normalized against the respective PMA groups. Data are from six independent experiments and are presented as median with interquartile range and 10–90 percentiles; mean “+”. Wilcoxon test with post hoc test. ** *p* < 0.005.

**Figure 4 cells-13-02111-f004:**
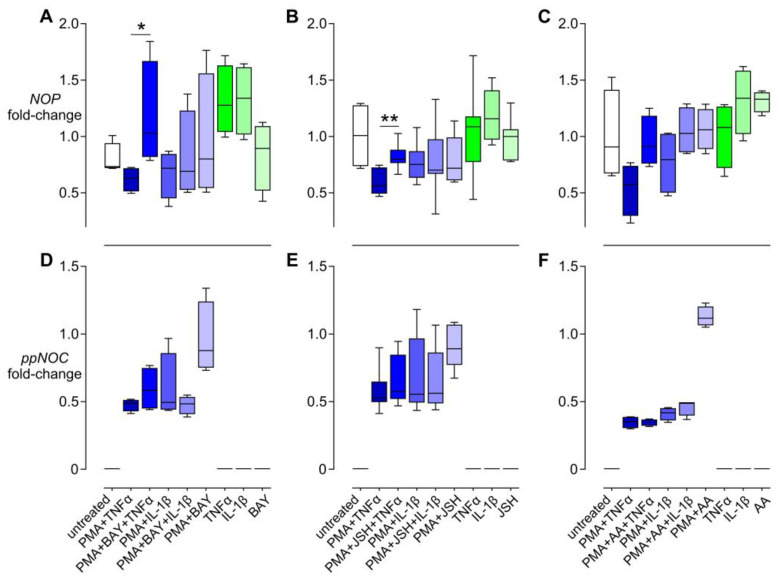
Effects of NFκB inhibitors on *NOP* and *ppNOC* mRNA expression. THP-1 cells were stimulated without/with PMA 5 ng/mL for 24 h, followed by treatment without/with one of the NFκB inhibitors (BAY 11-7082 (BAY, 100 nM), JSH-23 (JSH, 100 nM), and anacardic acid (AA, 100 nM)) for 1 h before exposure to TNF-α 10 ng/mL or IL-1β 10 ng/mL for 6 h (BAY and JSH) and for 12 h (AA). *NOP* (**A**–**C**) and *ppNOC* mRNA levels (**D**–**F**) are fold-change related to the corresponding PMA groups (controls). Median with interquartile range and 10–90 percentiles, experiment using BAY or AA, *n* = 4; experiment using JSH, *n* = 7. Mann–Whitney U test with post hoc test. * *p* < 0.05; ** *p* < 0.005.

**Figure 5 cells-13-02111-f005:**
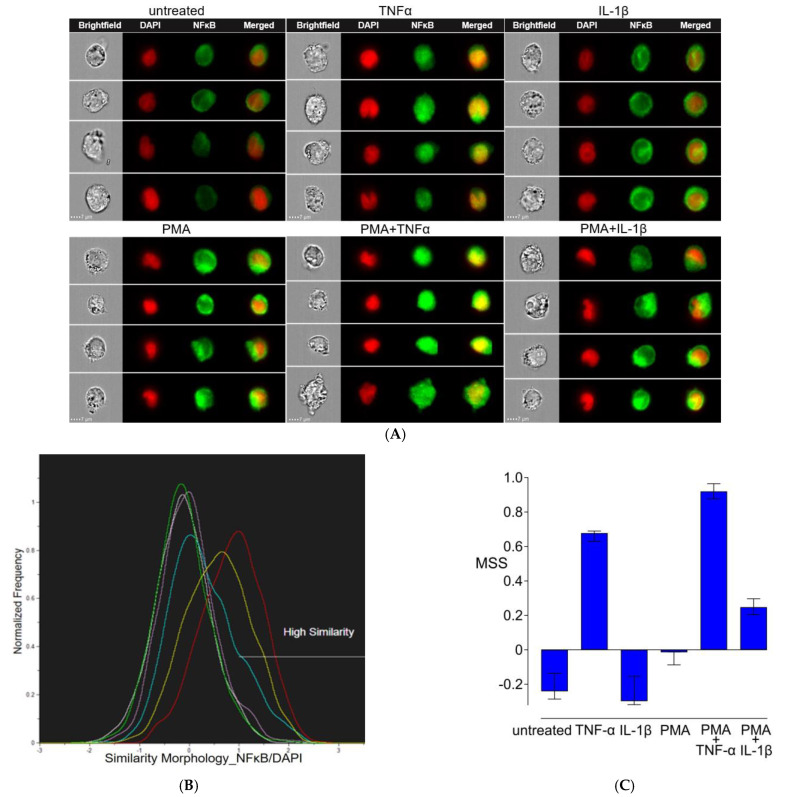
NFκB/p65 phosphorylation and nuclear translocation. THP-1 cells were treated without/with PMA 5 ng/mL for 24 h, followed by incubation with TNF-α 10 ng/mL or IL-1β 10 ng/mL or without additional cytokines for 1 h. (**A**) Composite images of representative cells (60×) in real time showing level of NFκB/p65 protein and its localization. From left to right in each panel, brightfield images of each cell, followed by the nuclear (red) and NFκB/p65 (green) images, and the merged image of the nucleus with NFκB/p65. The scale bar is 7 μm. (**B**) NFκB/p65 translocation into the nucleus, measured using similarity scores (SS). Histogram displaying varying SS in the cells of each group: untreated (white), TNF-α (yellow), IL-1β (green), PMA (purple), PMA + TNF-α (red) and PMA+ IL-1β (blue). (**C**) Median SS (MSS) for NFκB/p65 of the individual groups. Data are from three independent experiments and measures are expressed in median with interquartile range.

## Data Availability

The data in support of the findings of this study are available from the corresponding author upon reasonable request.

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
