# Peer review of "Nuclear Factor-κB Signaling Regulates the Nociceptin Receptor but Not Nociceptin Itself"

_cells, 2024, doi:10.3390/cells13242111_

Round 1
Reviewer 1 Report
Comments and Suggestions for Authors
This study aimed to elucidate the modulating effects of TNF-α and IL-1β and the regulation of NFκB on the nociceptin and the nociceptin receptor (NOP) in vitro using human monocytic THP-1 cells. The results showed that TNF-α, but not , IL-1β, decreases the PMA-induced up-regulation of NOP (mRNA). This effect was avoided if the cells were treated with NFκB inhibitors, suggesting that the impact of TNF-α depends on the activity of NFκB. In addition, TNF-α induced NFκB/p65 nuclear translocation in PMA-treated cells, but IL-1β did not. Suggesting a strong participation of TNF-α but not IL-1β in the effect of PMA's upregulating effects on nociceptin. NFκB seems to be involved in controlling the transcription of NOP.
I consider the theme relevant to understanding the transcriptional regulation of nociceptin and its receptor NOP. The research was conducted well, and the data obtained in a monocytic cellular line could contribute to understanding the mechanisms involved in the cells of the immune system. However, there are some points that the author should address to improve the quality of the manuscript.
1. The abstract contains a lot of information about methodological procedures; it must be redacted without the details of the hours of incubation or the groups analyzed and must be more general to indicate the techniques used to obtain the data. Also, the abstract must include the possible implications of their findings and the study's relevance, for example, in inflammation.
2. In the introduction section, it's necessary to give more details about the theme and make the writing more fluent; it is a little bit hard to find the main idea of the manuscript. For example, authors should bind that information to justify the research, like “LPS signaling involves toll-like receptors activation and the final activity of NFκB. NFκB is related to the transcriptional regulation of proinflammatory cytokines such as TNF-α and IL-1β, therefore is important to study the possible participation of NFκB and proinflammatory cytokines on the PMA-induced upregulation of nociceptin and NOP”.
3. In the results of the figure, there is no statistical analysis. Data are from two independent experiments, and measures are expressed in mean with range. I understand they performed this experiment to get the concentration they used for the rest of the experiments. However, there should be a higher number of independent experiments, at least to make a statistical analysis.
4. The discussion and the conclusion are consistent with the data obtained. However, authors should discuss the limitation of their study for example, even though TNF-α decreases the PMA-induced up-regulation of NOP, and possibly, this effect depends on the activity of NFκB, it is not necessarily the mechanism through LPS-induced up-regulation of NOP. Other signaling pathways could be involved.
Reviewer 2 Report
Comments and Suggestions for Authors
The study by Zhang et al. analyses the effect of nuclear factor kappaB on nociceptin and on the nociceptin receptor. The study is well designed, the methods are well described and the conclusion is supported by the results.
I just have a few questions.
It is important to clarify why non-parametric statistics were used throughout the studies.
The authors showed that TNF alfa promoted nuclear translocation of NF-kappaB, but this effect was not seen with IL-1beta. It would be beneficial to investigate the effect of both TNF alfa and IL-1beta on the phosphorylation and degradation of the IκB protein, which regulates the NF-kappaB pathway. This may help to elucidate the mechanism of action of the two cytokines.
